# DiffGlue: Diffusion-Aided Image Feature Matching

Shihua Zhang
Wuhan University
Wuhan, China
suhzhang001@gmail.com

Jiayi Ma*
Wuhan University
Wuhan, China
jyma2010@gmail.com

## Abstract

As one of the most fundamental computer vision problems, image feature matching aims to establish correct correspondences between two-view images. Existing studies enhance the descriptions of feature points with graph neural network (GNN), identifying correspondences with the predicted assignment matrix. However, this pipeline easily falls into a suboptimal result during training for the solution space is extremely complex, and is inaccessible to the prior that can guide the information propagation and network convergence. In this paper, we propose a novel method called DiffGlue that introduces the Diffusion Model into the sparse image feature matching framework. Concretely, based on the incrementally iterative diffusion and denoising processes, DiffGlue can be guided by the prior from the Diffusion Model and trained step by step on the optimization path, approaching the optimal solution progressively. Besides, it contains a special Assignment-Guided Attention as a bridge to merge the Diffusion Model and sparse image feature matching, which injects the inherent prior into GNN thereby ameliorating the message delivery. Extensive experiments reveal that DiffGlue converges faster and better, outperforming state-of-the-arts on several applications such as homography estimation, relative pose estimation, and visual localization. The code is available at https://github.com/SuhZhang/DiffGlue.

## CCS Concepts

• **Computing methodologies → Matching**.

## Keywords

Image feature matching, diffusion model, graph neural network

**ACM Reference Format:**

Shihua Zhang and Jiayi Ma. 2024. DiffGlue: Diffusion-Aided Image Feature Matching. In *Proceedings of the 32nd ACM International Conference on Multimedia (MM '24), October 28-November 1, 2024, Melbourne, VIC, Australia.* ACM, New York, NY, USA, 10 pages. https://doi.org/10.1145/3664647.3681069

## 1 Introduction

Image matching which aims to establish correct correspondences between different images from the same scene is a cardinal and

*Corresponding author.

critical procedure of many complex vision applications, such as panoramic stitching [7, 22], visual localization [44, 74], 3D reconstruction [16, 18], and neural rendering [68, 69]. A mature and effective pipeline of image matching begins with detecting and describing feature points in both images, where great efforts have been spent on applicable detectors and descriptors [13, 33, 65, 79]. Then matching these points with reference to their visual descriptions is applied by feature matching algorithms, identifying correspondences between the feature points of two-view images. This paper focuses on determining correct correspondences with existing feature points, better serving for subsequent tasks.

One of the most common ways to match feature points is searching the nearest neighbor (NN) through the similarity of descriptions. However, due to the inherent ambiguity of descriptions, these conventional methods struggle with large viewpoint changes, illumination variants, repetitive textures, and other complications [36]. Thanks to the strength of deep learning, SuperGlue [50] as a precursor builds a graph neural network (GNN) [54, 67] that transfers information among feature points to enhance their descriptions, and utilizes Sinkhorn algorithm [43, 57] to differentiably obtain the assignment matrix as the final matching results. Subsequently, the workflow of SuperGlue becomes a standard paradigm for image feature matching methods [9, 12, 32, 34, 75], and they derive the assignment matrix with Sinkhorn or Dual Softmax [60] during the training process so that learning a direct gradient decent to the ground truth (GT). However, the solution space of the assignment matrix is extremely unsmooth and complex, so the path to the optimal solution is tortuous and difficult to arrive by direct mapping only once in training, which is easy to trap in a local optimum, as shown in Figure 1(a). Moreover, SuperGlue has indicated the correlation between the real correspondences and the information propagation passageway of the feature points in different images, so that matching pairs should get greater information flow in the cross-attention. This phenomenon implies the importance of reliable correspondences or the assignment matrix as the prior to effectively transfer messages within the GNN, as shown in Figure 1(b). Thus, we are tempted to raise the following questions: **(i)** *Can we train the image feature matching model gradually on an optimization path so that the matching results approach the global optimum step by step?* **(ii)** *How to access the correspondence prior and inject it into GNN to ameliorate the information propagation?*

It is generally recognized that the Diffusion Model [11] can establish the mapping into a complex space [47], which may include the assignment matrix space as well. As shown in Figure 1(a), by adding Gaussian noise on GT incrementally, the Diffusion Model divides the optimization path into several parts during training instead of the common path with a direct gradient decent, then searches for the direction of gradient descent which guides the model to the optimal solution step by step with its inherent prior [20]. With the

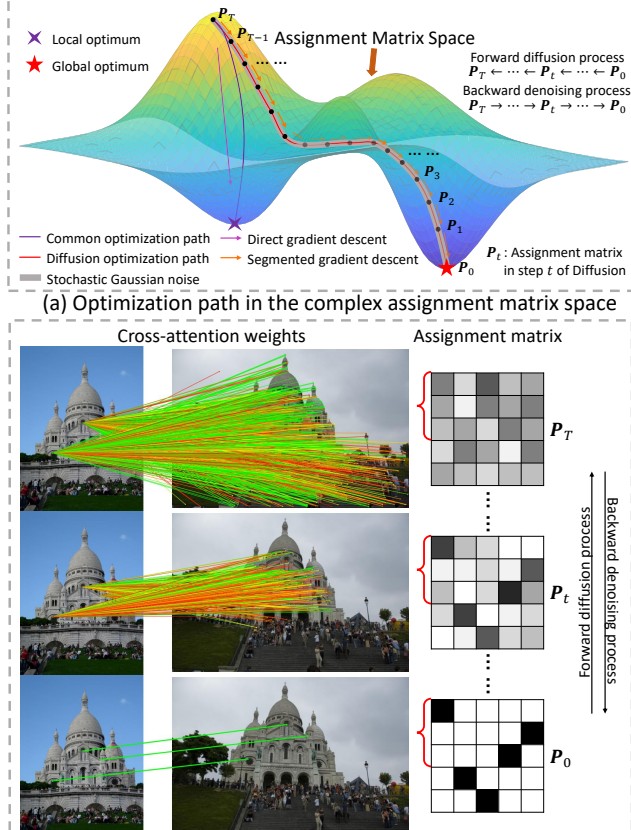

(a) Optimization path in the complex assignment matrix space

(b) Assignment matrix as the prior and the bridge

**Figure 1: Solutions to two problems in existing methods. In (a), we illustrate two different optimization paths in a pseudo assignment matrix space. $P_0$ indicates the GT and $P_t$ means the noised sampling in the diffusion pipeline. In (b), we plot the cross-attention weights of 3 feature points on the left, where connections with weights greater than 0.5 after min-max normalization are showcased, and mark low weights with red while high weights with green. Accordingly, we illustrate the GT $P_0$ together with its noised versions in the diffusion pipeline on the right and assume the first three rows indicate the results of the 3 points.**

segmented optimization path, the training data can be treated as a mixture of simple and difficult samples to improve model robustness and performance [46, 70], and the variety of noised training samples similar to data augmentation has been proved to avoid local minimal [56], achieving a better convergence. This is precisely an effective manner to solve the problem **(i)**. Besides, as shown in Figure 1(b), the attention weights are always consistent with the assignment matrix, and its gradual changes are very similar to the diffusion pipeline. It reveals that the assignment matrix obtained by the Diffusion Model in the former step can act as the prior required by the latter step, and can directly guide the information propagation in the GNN to solve the problem **(ii)**. For these reasons, we naturally consider introducing the Diffusion Model into image

feature matching to solve the above questions and facilitate better model convergence and more effective information propagation.

However, most of the existing Diffusion Models are used for tasks where the input is regular images [14], and the common model structures are not suitable for image feature matching with sparse feature points as input. Even though some recent works have attempted to use the Diffusion Model on point cloud tasks, the goal is to generate recognition results [42] or relative pose [23], and could not provide instructions on how to connect the Diffusion Model and the assignment matrix of correspondences. Inspired by SuperGlue [50] that the cross-attention map is positively correlated with the assignment matrix as shown in Figure 1(b), we design a special Assignment-Guided Attention that imitates cross-attention but substitutes the assignment matrix for the attention map to inject correspondence prior into the GNN and make the assignment matrix a bridge between image feature matching network and Diffusion Model framework. Following this train of thought, we propose DiffGlue, a Diffusion-aided image feature matching algorithm, to perform the process of the Diffusion Model with the help of Assignment-Guided Attention, achieving accurate and robust matching results. We also apply the sampling acceleration technique DDIM [58] to ensure the matching efficiency of DiffGlue.

In summary, our main contributions are as follows:

- We propose DiffGlue, that introduces the Diffusion Model into sparse image feature matching, which helps to find the optimal solution incrementally in the assignment matrix space guided by the inherent prior of the Diffusion Model.
- We choose the assignment matrix as the bridge to connect the Diffusion Model and sparse correspondences, and design an Assignment-Guided Attention to inject correspondence prior into the GNN, hence guiding the information propagation.
- The evaluation is extensively conducted across a variety of practical tasks, state-of-the-art results are reported that demonstrate the superiority of DiffGlue. And the effect of each component is thoroughly studied.

## 2 Related Work

### 2.1 Classic Image Feature Matching

Traditionally, image feature matching starts with **feature detection and description**. Handcrafted detector-descriptors include gradient statistic-based methods [5, 33] and intensity comparison-based methods [8, 48]. Deep methods are developed to extract more distinct feature points and discriminative descriptions [13, 35, 65, 79]. Then with the similarity of the descriptions, one can **match the feature points** to obtain a coarse correspondence set with the NN, the MNN, or the distance ratio test method [26]. But due to the limited discriminability of the descriptor, numerous outliers exist in the coarse set. Thus, **outlier rejection** is necessary to retain inliers and remove outliers, *e.g.*, the traditional methods [25, 37, 38] and the learning-based ones [31, 76–78]. After deriving clean correspondences, **pose estimation** is typically employed for subsequent vision tasks. In addition to the direct regression of pose with Direct Linear Transform (DLT), the robust estimator RANSAC [15] is commonly used, together with its variants [3, 4]. However, unlike this phased pipeline, we directly determine correspondences from

feature points and descriptions in an end-to-end manner, matching feature points and removing outliers jointly.

## 2.2 End-to-End Image Feature Matching

SuperGlue [50] as a pioneering first accomplishes accurate end-to-end image feature matching, relying on GNN's powerful intra- and inter-graph information transfer capabilities [66, 67]. Based on this seminal work, some researchers further enhance the network capacities and optimize some details, *e.g.*, KeyGNN [27] focuses on the information flow on structure-important and texture-rich feature points, ParaFormer [34] designs a novel parallel attention and U-type GNN, IMP [75] emphasis the role of positional information in feature matching, and ResMatch [12] facilitates self- and cross-attention by adding relative position and the similarity of descriptions. While others attempt to speed up the paradigm, *e.g.*, SGMNet [9] introduces seeding strategy and reduces the computation complexity from $O(N^2)$ to $O(N)$ where $N$ is the number of feature points, and LightGlue [32] proposes an early termination scheme and improves the cross-attention. However, all methods employ either Sinkhorn [57] or Dual Softmax [60] to obtain the assignment matrix to identify correspondences, not considering the complexity of the solution space and the accuracy of the convergence point during the training process. Although SuperGlue keenly perceives the importance of the matching prior in the message passing of GNN [50], existing methods do not attempt to inject it into the network. In this paper, we try to learn the optimization path in the assignment matrix space with Diffusion Model [20, 41] progressively, accelerating the convergence to a global optimum, and to inject correspondences prior from the Diffusion Model into GNN with a specially designed attention. Moreover, LoFTR [60] and its successors [17, 40] build correspondences with dense features, therefore often suffer from inefficiencies of the feature extractor. We believe the proposed diffusion-based framework still deserves to be studied since it can be applied similarly to LoFTR which also utilizes GNN and Dual Softmax to match accurately.

## 2.3 Diffusion Model in Geometric Alignment

Diffusion Model, which gradually perturbs the input data over several steps by adding Gaussian noise and recovers the original data by learning to gradually reverse the diffusion process step by step [11], now not only shines in image generation [47] or discriminative tasks [10, 21], but also shows excellent potential in geometric alignment. For 2D image alignment, the conditions usually are different images, guiding the Diffusion Model to generate optical flow [53] or dense warping [39]. There are also methods that attempt to produce the transformed image [29] or regress the relative pose directly [69]. But all of them handle regular images, not applicable to image feature matching where the conditions are sparse feature points. In 3D point alignment, DiffusionReg [23] attempts to deal with sparse points and embeds the Diffusion Model in the SE(3) space. It predicts 6D object pose finally instead of the correspondences of points. So to the best of our knowledge, there is not yet a Diffusion Model for image feature matching. The most similar concurrent works are [71, 72], which utilize the diffusion model to learn the matching matrix in point cloud registration, guiding the learning of the rigid transformation and gradually correcting

the alignment errors. In contrast, our approach focuses on leveraging the matching prior in the diffusion model, and designs the Assignment-Guided Attention module to inject this prior into the sparse image matching pipeline to improve the performance.

## 3 Preliminaries

### 3.1 Revisiting the Diffusion Model

Diffusion Model, as a class of probabilistic generative models, aims to build a complex mapping function by learning to reverse a noising process that gradually degrades the training data. It involves two phases: a forward diffusion process that gradually converts the data $x$ into Gaussian noise in $T \in \mathbb{N}$ steps, and a backward denoising process that removes the noise progressively with a learnable denoising network $\mathcal{D}_\theta(\cdot)$.

According to DDPM [20], the forward process can be defined as a Markovian process:

$$p(x_t|x_{t-1}) = \mathcal{N}(x_t; \sqrt{1-\beta_t}x_{t-1}, \beta_t \mathbf{I}), \quad \forall t \in \{1, \ldots, T\}, \quad (1)$$

where $T$ is the number of diffusion steps, $x_0 \sim p(x_0)$ means an uncontaminated sample with $p(x_0)$ being the data density, $x_1, \ldots, x_T$ are the noised data and satisfy $p(x_T) \approx \mathcal{N}(\mathbf{0}, \mathbf{I})$ finally, $\beta_1, \ldots, \beta_T \in [0, 1)$ are hyperparameters of the variance schedule. The recursive formulation Eq. (1) allows a direct sampling of $x_t$ with $x_0$:

$$p(x_t|x_0) = \mathcal{N}(x_t; \sqrt{\hat{\beta}_t}x_0, (1-\hat{\beta}_t)\mathbf{I}), \quad (2)$$

where $\hat{\beta}_t = \prod_{i=1}^t \alpha_i, \alpha_t = 1 - \beta_t$. Then the training is performed by minimizing a variational lower-bound of the negative log-likelihood:

$$\mathcal{L}_{\text{vlb}} = -\log p_\theta(x_0|x_1) + \text{KL}(p(x_T|x_0)\|p(x_T))$$
$$+ \sum_{t>1} \text{KL}(p(x_{t-1}|x_t, x_0)\|p_\theta(x_{t-1}|x_t)), \quad (3)$$

where $\text{KL}(\cdot\|\cdot)$ denotes the Kullback-Leibler divergence. The posterior $p(x_{t-1}|x_t, x_0)$ here can be proven as a Gaussian distribution with the Bayes rule [20] based on Eqs. (1) and (2):

$$p(x_{t-1}|x_t, x_0) = \frac{p(x_t|x_{t-1})p(x_{t-1}|x_0)}{p(x_t|x_0)} \propto \mathcal{N}(x_{t-1}; \mu_t(x_t, x_0), \tilde{\beta}_t \mathbf{I}), \quad (4)$$

where $\tilde{\beta}_t = \frac{1-\hat{\beta}_{t-1}}{1-\hat{\beta}_t}\beta_t$. With [28], the solution of Eq. (3) is:

$$\arg\min_\theta \text{KL}(p(x_{t-1}|x_t, x_0)\|p_\theta(x_{t-1}|x_t)) = \arg\min_\theta \|\mathcal{D}_\theta(x_t) - x_0\|. \quad (5)$$

Thus, the optimization is equivalent to learning a denoising network $\mathcal{D}_\theta(\cdot)$ to predict the GT $x_0$ from $x_t$. And with $p_\theta(x_{t-1}|x_t) \approx p(x_{t-1}|x_t, x_0 = \mathcal{D}_\theta(x_t))$ we acquire the target from any given $x_T$.

### 3.2 Problem Formulation

Given an image, local feature coordinates $c_i \in \mathbb{R}^2$ and descriptions $d_i \in \mathbb{R}^{C_d}$ can be respectively obtained by an off-the-shelf feature extractor, where $i = 1, \ldots, M$ means the $i$-th point and $C_d$ is the channel length of the description. With $C_A = \{c_{A,i}\}, D_A = \{d_{A,i}\}, i = 1, \ldots, M$ of the source image $A$ and $C_B = \{c_{B,i}\}, D_B = \{d_{B,i}\}, i = 1, \ldots, N$ of the target image $B$, correct correspondences $\mathcal{M}_{A,B}$ can be established:

$$\mathcal{M}_{A,B} = \{(i, j)|\|\mathcal{T}(c_{A,i}) - \mathcal{T}(c_{B,j})\|_2 \leqslant \epsilon\}, \quad (6)$$

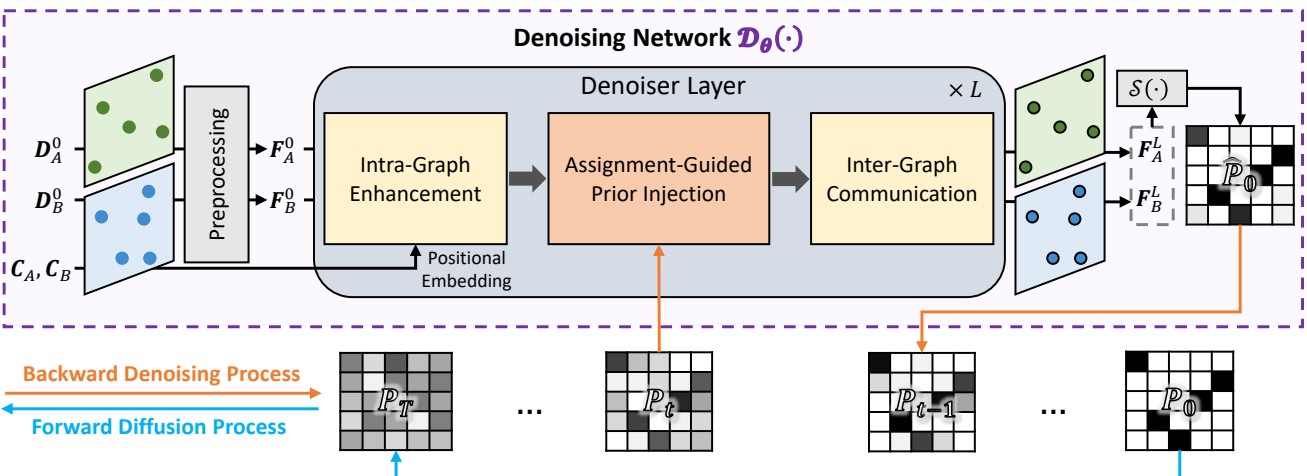

**Figure 2: Framework of DiffGlue and architecture of the denoising network $\mathcal{D}_\theta$. The input of $\mathcal{D}_\theta$ in step $t$ is the correspondence prior $P_t$ from the former step, with descriptions $D_A, D_B$ and coordinates $C_A, C_B$ as conditions. After preprocessing, features $F_A^0, F_B^0$ converted from descriptions are fed into $L$ stacked Denoiser layer, each layer includes three modules, *i.e.*, intra-graph enhancement based on self-attention, assignment-guided prior injection based on the proposed Assignment-Guided Attention, and inter-graph communication based on cross-attention, and outputs updated features $F_A^L, F_B^L$. With these new features, the denoising network predicts a pseudo assignment matrix $\hat{P}_0$ by $\mathcal{S}(\cdot)$ in Eq. (7), then calculates $P_{t-1}$ for next step.**

where $\mathcal{T}(\cdot)$ represents a function that converts image coordinates to world coordinates. For learnable image feature matching methods, they enhance local features $D$ to $F$ with the designed network, then calculate the assignment matrix $P \in \mathbb{R}^{M \times N}$ with Sinkhorn [57] or Dual Softmax [60] projection function $\mathcal{S}(\cdot)$ to obtain $\mathcal{M}_{A,B}$, we choose Dual Softmax here for example:

$$P = \mathcal{S}(F_A(F_B)^T) = \text{Softmax}(F_A(F_B)^T) \odot \text{Softmax}(F_B(F_A)^T)^T,$$ 
(7)

$$\mathcal{M}_{A,B} = \text{Index}(P \geq \delta),$$ 
(8)

where $\odot$ is the Hadamard production, $\text{Index}(\cdot)$ finds index $(i, j)$ that satisfies the certain condition. In this paper, we predict the assignment matrix with the Diffusion Model as $P_0 = \mathcal{D}_\theta(P_t)$ in each time step $t$ for better convergence during training.

## 4 Methodology

The main concept of DiffGlue is introducing the Diffusion Model into learnable image feature matching, enabling better convergence of the network. To apply Diffusion Model appropriately for the sparse feature points as input, we propose a special Assignment-Guided Attention then design a GNN-based denoising network. We will present all of these together with the loss function and implementation details in this chapter.

### 4.1 Diffusion Model for Image Feature Matching

We construct a Diffusion Model, including a forward diffusion process and a backward denoising process, to generate the assignment matrix $P \in [0, 1]^{M \times N}$ of two feature point sets from different images $A$ and $B$. Refer to the revisit of Diffusion Model in Sec. 3.1, we detail our implementations on image feature matching.

*4.1.1 Forward Diffusion Process.* The forward diffusion Markovian process in Eq. (1) could also be constructed on the GT assignment

matrix $P_0$ in the complex assignment solution space as:

$$p(P_t|P_{t-1}) = \mathcal{N}(P_t; \sqrt{1 - \beta_t}P_{t-1}, \beta_t \mathbf{I}), \quad \forall t \in \{1, \ldots, T\}. \quad (9)$$

Then according to Eq. (2), $P_t$ has a direct closed form similarly:

$$p(P_t|P_0) = \mathcal{N}(P_t; \sqrt{\hat{\beta}_t}P_0, (1 - \hat{\beta}_t)\mathbf{I}), \quad (10)$$

which derives a sequence of noised samplings $P_t$ from $P_0$, and satisfies $P_T \sim \mathcal{N}(\mathbf{0}, \mathbf{I})$. However, the sampled $P_t$ may be beyond the assignment matrix space $\{P \in [0, 1]^{M \times N}\}$, so we use shifting and clipping techniques similar to image generation [14, 47] where RGB value should not exceed [0, 255], ensuring $P_t \in [-1, 1]^{M \times N}$ during the forward and backward processes.

*4.1.2 Backward Denoising Process.* To guarantee the calculability of (3), we rewrite the posterior in Eq. (4):

$$p(P_{t-1}|P_t, P_0) \propto \mathcal{N}(P_{t-1}; \mu_t(P_t, P_0), \tilde{\beta}_t \mathbf{I}). \quad (11)$$

Hence once obtaining the predicted target $\hat{P}_0$ with the denoising network $\mathcal{D}_\theta(\cdot)$, we can derive the input of next denoising step $P_{t-1}$ by substituting $P_0$ with $\hat{P}_0 = \mathcal{D}_\theta(P_t)$:

$$p(P_{t-1}|P_t, P_0 = \hat{P}_0) = p(P_{t-1}|P_t, P_0 = \mathcal{D}_\theta(P_t))$$
$$\propto \mathcal{N}(P_{t-1}; \mu_t(P_t, \mathcal{D}_\theta(P_t)), \tilde{\beta}_t \mathbf{I}). \quad (12)$$

After $T$ steps [20] or less steps [58], we can get the final assignment matrix $\hat{P}$ and obtain the correspondences. Overall, the forward diffusion process and the backward denoising process are summarized in Figure 2. The details of $\mathcal{D}_\theta(\cdot)$ are introduced in Sec. 4.2.

*4.1.3 Optimization Loss.* With Eq. (5), we maintain only the last term as a simple loss function referring to [6]. So, during training, $\mathcal{D}_\theta(\cdot)$ is supervised with the following denoising loss:

$$\mathcal{L}_{\text{diff}} = E_{t \sim [1,T], P_t \sim p(P_t|P_0)} \|\mathcal{D}_\theta(P_t) - P_0\|^2. \quad (13)$$

For $P_t$ is sampled from denoised distribution, the starting points of each training session are spread out over the optimization path. And together with the wider exploration of the solution space by stochastic Gaussian noise, the diffusion-based model will search for the direction of gradient descent in the complex assignment matrix space, speeding up model convergence and preventing the algorithm from falling into local optimum.

## 4.2 Denoising Network

The denoising network $\mathcal{D}_\theta(\cdot)$ mainly contains three modules converted from the GNN unit: intra-graph enhancement based on self-attention, correspondence prior injection based on Assignment-Guided Attention, and inter-graph communication based on cross-attention, respectively. These modules compose the Denoiser Layer, which is stacked $L$ times in $\mathcal{D}_\theta(\cdot)$. As shown in Figure 2, the input of $\mathcal{D}_\theta(\cdot)$ is the assignment matrix $P_t$ from the former step together with descriptions $D_A, D_B$ and coordinates $C_A, C_B$ as conditions. After denoising, $\mathcal{D}_\theta(\cdot)$ generates a pseudo assignment matrix $\hat{P}_0$, then the input of next step $P_{t-1}$ can be obtained by Eq. (12). In this section, we introduce the architecture of $\mathcal{D}_\theta(\cdot)$ exhaustively.

*4.2.1 Preprocessing.* For each timestamp $t \sim [1, T]$, we first encode it with embedding function $\mathcal{E}_{\text{time}}(\cdot)$ with reference to [14]:

$$\tau_t = \mathcal{E}_{\text{time}}(t), \quad t \sim [1, T], \tau_t \in \mathbb{R}^C, \tag{14}$$

where $C$ is the channel length of the features in $\mathcal{D}_\theta(\cdot)$. To align the dimension of the descriptions $D_I, I \in \{A, B\}$ with the network features, we project $D_I$ to the input features $F_I^0$ of the 1-st layer with a multi-layer perception (MLP):

$$F_I^0 = \mathcal{E}_{\text{desc}}(D_I), \quad I \in \{A, B\}, D_I \in \mathbb{R}^{C_d}, F_I^0 \in \mathbb{R}^C. \tag{15}$$

*4.2.2 GNN Unit.* The initial $F_I^0$ is then updated gradually from $F_I^1$ to $F_I^L$ where $F_I^\ell$ is obtained by the $\ell$-th layer GNN unit $\mathcal{G}^\ell(\cdot)$:

$$
\begin{aligned}
F_I^\ell &= \mathcal{G}^\ell(F_I^{\ell-1}, F_J^{\ell-1}) \\
&= F_I^{\ell-1} + \text{FFN}(F_I^{\ell-1} \| \mathcal{A}(F_I^{\ell-1}, F_J^{\ell-1})),
\end{aligned}
\tag{16}
$$

where $\|$ denotes concatenating by channels, FFN($\cdot$) means feedforward network (FFN) that compacts the result of concatenation so that the channel length is equal to $C$, $I, J \in \{A, B\}$, and $\ell = 1, \ldots, L$. It is the difference in function $\mathcal{A}(\cdot)$ that produces the three modules mentioned above. Details of each can be seen subsequently.

*4.2.3 Intra-Graph Enhancement.* Similar to [50, 60], we construct $\mathcal{G}_{\text{self}}(\cdot)$ based on self-attention [66] for all the feature points within each image $I$ in order to enhance the intra-graph presentations:

$$F_{I,\text{self}}^{\ell-1} = \mathcal{G}_{\text{self}}^\ell(F_I^{\ell-1}, F_I^{\ell-1}). \tag{17}$$

And here we define $\mathcal{A}(\cdot)$ in Eq. (16) as:

$$
\begin{aligned}
\mathcal{A}(F_I, F_I) &= \text{Softmax}\left(\frac{(W_Q F_I)(W_K F_I)^T}{\sqrt{C}}\right) W_V F_I \\
&= \text{Softmax}\left(\frac{Q_I K_I^T}{\sqrt{C}}\right) V_I = \text{Softmax}(\alpha_{II}) V_I,
\end{aligned}
\tag{18}
$$

where $W_Q, W_K, W_V$ are learnable weights, $Q_I, K_I, V_I \in \mathbb{R}^{M \times C}$ with $M$ being the number of feature points in image $I$. Besides, we

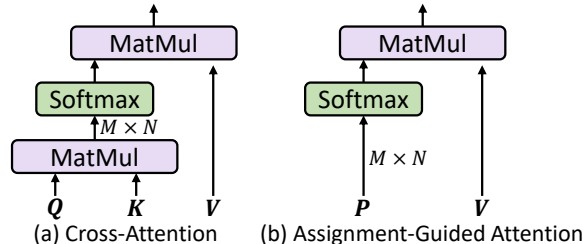

(a) Cross-Attention    (b) Assignment-Guided Attention

**Figure 3: Structure comparisons between cross-attention and the proposed Assignment-Guided Attention, 'MatMul' means matrix multiplication.**

add relative positional embedding [32, 73] for each $\alpha_{i,j} \in \alpha_{II}$:

$$\alpha_{i,j} = \frac{q_i R(c_j - c_i) k_j^T}{\sqrt{C}}, \tag{19}$$

$R(\cdot) \in \mathbb{R}^{C \times C}$ is a rotary encoding introduced thoroughly in [59].

*4.2.4 Assignment-Guided Prior Injection.* At timestamp $t$, correspondence prior $P_t$ can be obtained from the former step. Before injecting it into GNN, we first embed the timestamp information:

$$F_{I,\text{embed}}^{\ell-1} = F_{I,\text{self}}^{\ell-1} + \tau_t. \tag{20}$$

Then we inject the prior $P_t$ with $\mathcal{G}_{\text{assign}}(\cdot)$ based on the designed Assignment-Guided Attention. Still similar to Eq. (16), we choose features in image $A$ as an example:

$$F_{A,\text{assign}}^{\ell-1} = \mathcal{G}_{\text{assign}}^\ell(F_{A,\text{embed}}^{\ell-1}, F_{B,\text{embed}}^{\ell-1}, P_t). \tag{21}$$

Here, the function $\mathcal{A}(\cdot)$ is rewritten accordingly by replacing $\alpha_{II}$ in Eq. (18) with $\alpha_{AB} = P_t$ as:

$$\mathcal{A}(F_A, F_B) = \text{Softmax}(\alpha_{AB}) V_B = \text{Softmax}(P_t) V_B. \tag{22}$$

And $F_{B,\text{assign}}^{\ell-1}$ can be also obtained easily with:

$$\mathcal{A}(F_B, F_A) = \text{Softmax}(\alpha_{BA}) V_A = \text{Softmax}(P_t^T) V_A. \tag{23}$$

Eqs. (22) and (23) are so-called the Assignment-Guided Attention. Actually, the proposed attention is more like a degenerate version of cross-attention, as shown in Figure 3. It builds a lightweight structure to connect the more relative features between two images with the correspondence prior from the Diffusion Model, and important messages will be delivered among the feature points through these strong links, thus further improving the matching quality of the new features. During the training, the injected prior information guides the denoiser $\mathcal{D}_\theta(\cdot)$ converges step by step, so that conquers the difficulty of trapping in the local optimum.

*4.2.5 Inter-Graph Communication.* Although it has been discussed that the Assignment-Guided Attention could achieve information interaction across images to some extent using the correspondence prior, we still retain the inter-graph communication based on cross-attention to ensure the capability of $\mathcal{D}_\theta(\cdot)$ [50]. First, we merge the acquired assignment-guided features $F_{I,\text{assign}}^{\ell-1}$ with the input features of this layer $F_I^{\ell-1}$ with an MLP $\mathcal{E}_{\text{merge}}(\cdot)$:

$$F_{I,\text{merge}}^{\ell-1} = \mathcal{E}_{\text{merge}}(F_{I,\text{assign}}^{\ell-1} \| F_I^{\ell-1}). \tag{24}$$

Then using $\mathcal{G}_{\text{cross}}(\cdot)$, we perform the inter-graph communication. Here we still choose features in image $A$ as an example:

$$F_{A,\text{cross}}^{\ell-1} = \mathcal{G}_{\text{cross}}^{\ell}(F_{A,\text{merge}}^{\ell-1}, F_{B,\text{merge}}^{\ell-1}). \tag{25}$$

The function $\mathcal{A}(\cdot)$ here is rewritten as:

$$\mathcal{A}(F_A, F_B) = \text{Softmax}\left(\frac{Q_A K_B^{T}}{\sqrt{C}}\right)V_B = \text{Softmax}(\boldsymbol{\alpha}_{AB})V_B. \tag{26}$$

And for image $B$:

$$\mathcal{A}(F_B, F_A) = \text{Softmax}(\boldsymbol{\alpha}_{AB}{}^{T})V_A. \tag{27}$$

Then the output of layer $\ell$ and the input of layer $\ell+1$ is received as $F^{\ell} = F_{I,\text{cross}}^{\ell-1}$, until the last layer yielding $F_I^L$. Then the pseudo target assignment matrix $\hat{P}_0$ can be calculated according to Eq. (7).

## 4.3 Loss Function

In addition to the diffusion loss mentioned in Eq. (13), we still retain the matching loss similar to [32, 60]. Specifically, we predict the matchability score:

$$\boldsymbol{\sigma}_I = \text{Sigmoid}(\mathcal{E}(F_I^L)) \in [0,1]^M, \tag{28}$$

where $\mathcal{E}(\cdot)$ is an MLP that projects $F_I^L$ to only one dimension. Then the augmented assignment matrix $\tilde{P}$ is derived by:

$$\tilde{P} = \boldsymbol{\sigma}_A{}^{T}\boldsymbol{\sigma}_B \odot \hat{P}. \tag{29}$$

This equation can be generalized to each layer to get $\tilde{P}^{\ell}$ where $\ell = 1, \ldots, L$. Hence, the matching loss function is:

$$\mathcal{L}_{\text{match}} = -\frac{1}{L}\sum_{\ell}\left(\frac{1}{|\mathcal{M}_{\text{gt}}|}\sum_{(i,j)\in\mathcal{M}_{\text{gt}}}\log\tilde{P}_{ij}^{\ell}\right.$$
$$\left. +\frac{1}{2|\bar{\mathcal{I}}|}\sum_{i\in\bar{\mathcal{I}}}\log\left(1-\sigma_{A,i}^{\ell}\right) + \frac{1}{2|\bar{\mathcal{J}}|}\sum_{j\in\bar{\mathcal{J}}}\log\left(1-\sigma_{B,j}^{\ell}\right)\right), \tag{30}$$

where $\mathcal{M}$ is the GT correspondence with a low reprojection error, and $\bar{\mathcal{I}}, \bar{\mathcal{J}}$ are unmatchable points. Finally, according to Eqs. (13) and (30), the total loss of DiffGlue is:

$$\mathcal{L} = \mathcal{L}_{\text{match}} + \lambda\mathcal{L}_{\text{diff}}, \tag{31}$$

where $\lambda$ is a hyper-parameter to balance different loss functions.

## 4.4 Implementation Details

We stack the Denoiser Layer in $\mathcal{D}_{\theta}(\cdot)$ 9 times (i.e. $L = 9$). The head number of attention in Eqs. (17) and (25) is 4, and the channel length $C$ of features in $\mathcal{D}_{\theta}(\cdot)$ is 256. According to the strategy in [32, 50], we adopt the same datasets for two-stage training, Oxford and Paris [45] for synthetic homography pre-training, MegaDepth [30] for fine-tuning. Specifically, in the first stage, we resize images to $640 \times 480$, extract 512/1024 feature points with SuperPoint [13]/ALIKED [79]. Batch size 64 while the learning rate is 0.0001, we multiply the learning rate by 0.8 each epoch after 20 epochs, and stop the training after 40 epochs. We set $\lambda = 1000$ in Eq. (31). In the second stage, images are resized to $1024 \times 1024$ with zero-padding, feature points are extracted up to 2048. Batch size is 32 while the learning rate is 0.0001 for 20 epochs then decayed by a factor of 10 over 10 epochs until 40 epochs. $\lambda$ is set as 1. The total step number $T$ of the Diffusion Model is 4096. But for inference, we

**Table 1: Homography estimation on HPatches [2].**

| Feature+Matcher | | Acc. | | AUC | | | |
|---|---|---|---|---|---|---|---|
| | | | | DLT | | RANSAC [15] | |
| | | @1px | @3px | @1px | @3px | @1px | @3px |
| SP [13] | MNN | 26.8 | 74.7 | 0.37 | 1.90 | 32.05 | 51.06 |
| | SuperGlue [50] | 32.7 | 92.7 | 32.08 | 64.96 | 33.48 | 56.16 |
| | SGMNet [9] | 31.9 | 89.0 | 17.93 | 48.39 | 32.27 | 53.86 |
| | ResMatch [12] | 31.1 | 87.3 | 31.23 | 64.53 | 32.28 | 55.05 |
| | IMP [75] | 31.2 | 87.7 | 24.02 | 54.32 | 33.49 | 56.06 |
| | LightGlue [32] | 33.6 | 94.6 | 34.67 | 66.36 | 34.87 | 56.36 |
| | DiffGlue (Ours) | 33.8 | 95.6 | 35.72 | 67.34 | 35.05 | 57.62 |
| ALIKED [79] | MNN | 54.4 | 86.2 | 4.51 | 16.02 | 18.85 | 50.53 |
| | LightGlue | 63.8 | 97.9 | 19.01 | 51.77 | 31.12 | 64.43 |
| | DiffGlue (Ours) | 64.2 | 98.5 | 19.97 | 53.07 | 32.79 | 65.68 |

use DDIM technique [58] and sample only 2 steps with $\delta = 0.1$ in Eq. (8). For practical real-time applications, we use predictions from the first step directly, which has yielded remarkable results. All training processes are conducted on two NVIDIA RTX3090 GPUs. The testing in the following is performed with a single GPU while the random seed is set as 0 to obtain stable results.

## 5 Experiments

We evaluate DiffGlue on homography estimation, relative pose estimation and visual localization. Additionally, we analyze the convergence and computational usage of DiffGlue, then discuss the effectiveness of the Diffusion-based framework and the Assignment-Guided Attention by conducting ablation studies.

## 5.1 Homography Estimation

Homography estimation is a basic task in computer vision to find a linear image-to-image map in homogeneous space, and we perform this experiment on HPatches benchmark [2], referring the settings in [60]. We first resize all images so that their smaller dimension is 480 pixels, then extract up to 1024 feature points with both SuperPoint (SP) [13] and ALIKED [79] for each image, identify correspondences with image feature matching methods, and finally estimate the homography transformation with both non-robust estimator (i.e. DLT) and robust estimator (i.e. RANSAC [15]). For each image pair, we try to classify a correspondence to be right or not and calculate the accuracy (i.e. **Acc.**) at 1 and 3 pixels as the same as [32], and we also adopt the mean reprojection error of the four image corners according to [13] and report the area under the cumulative error curve (**AUC**) at multiple thresholds (1 and 3 pixels). We choose the MNN as a baseline, comparing DiffGlue with several image feature matching methods, i.e., SuperGlue [50], SGMNet [9], ResMatch [12], IMP [75], and LightGlue [32]. Results are shown in Table 1. DiffGlue outperforms all other methods.

## 5.2 Relative Pose Estimation

Recovering camera relative pose (rotation and translation) from two-view images is a key step in many vision applications. The accuracy of relative pose estimation can reflect the performance of image feature matching methods. Following the settings in [32, 50, 60], we choose MegaDepth-1500 [30] and YFCC100M [62] datasets for experiments. Concretely, we first resize the images so that their

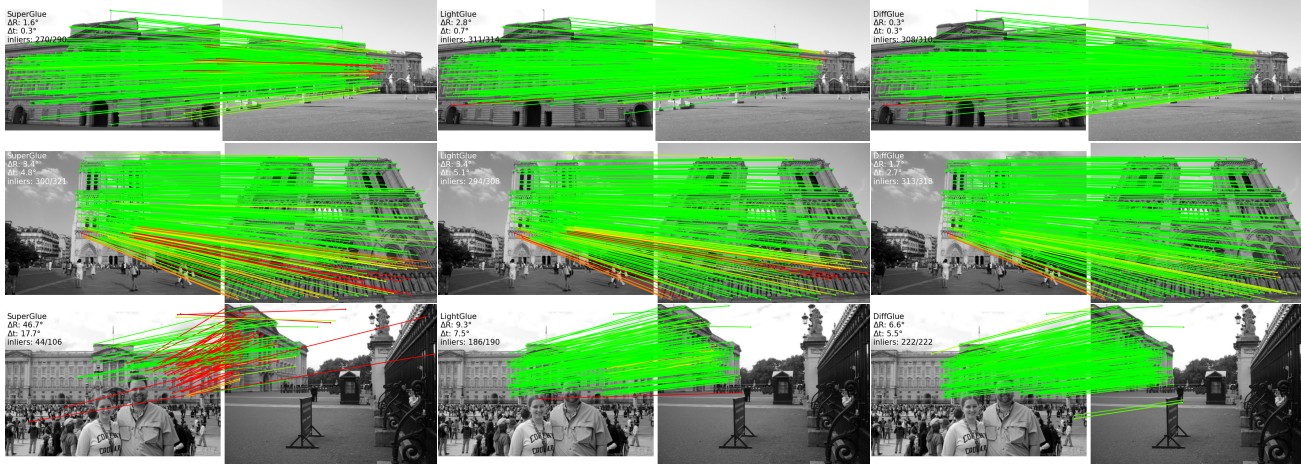

**Figure 4: Qualitative illustration of outlier rejection. Mark false matches with red while correct matches with green. The relative pose estimation results are provided in the top left corner. Zoom in for better visualization.**

**Table 2: Relative pose estimation on MegaDepth-1500 [30].**

| Feature+Matcher | | AUC | | | | | |
|---|---|---|---|---|---|---|---|
| | | DLT | | RANSAC [15] | | MAGSAC++ [4] | |
| | | @5° | @10° | @5° | @10° | @5° | @10° |
| SP [13] | MNN | 0.10 | 0.21 | 29.31 | 44.85 | 27.73 | 42.22 |
| | SuperGlue [50] | 32.34 | 47.68 | 48.44 | 65.70 | 60.88 | 75.25 |
| | SGMNet [9] | 4.78 | 11.22 | 39.95 | 58.49 | 47.63 | 64.56 |
| | ResMatch [12] | 26.38 | 41.01 | 43.86 | 61.37 | 54.26 | 69.59 |
| | IMP [75] | 32.96 | 48.59 | 44.94 | 62.45 | 56.86 | 71.87 |
| | LightGlue [32] | 39.14 | 55.22 | 47.78 | 65.51 | 61.45 | 75.29 |
| | DiffGlue (Ours) | **43.21** | **59.13** | **50.21** | **67.30** | **63.06** | **76.61** |
| ALIKED [79] | MNN | 0.55 | 1.94 | 47.71 | 62.67 | 44.62 | 59.87 |
| | LightGlue | 45.03 | 60.40 | 50.85 | 67.39 | 64.17 | 76.83 |
| | DiffGlue (Ours) | **47.38** | **63.05** | **51.31** | **67.99** | **64.44** | **77.39** |

**Table 3: Relative pose estimation on YFCC100M [62].**

| Feature+Matcher | | AUC | | | | | |
|---|---|---|---|---|---|---|---|
| | | DLT | | RANSAC [15] | | MAGSAC++ [4] | |
| | | @5° | @10° | @5° | @10° | @5° | @10° |
| SP [13] | MNN | 0.01 | 0.11 | 15.72 | 29.66 | 15.6 | 29.19 |
| | SuperGlue [50] | 19.06 | 33.07 | 39.47 | 59.75 | 47.77 | 66.94 |
| | SGMNet [9] | 9.84 | 19.85 | 34.22 | 54.50 | 35.26 | 55.75 |
| | ResMatch [12] | 18.10 | 31.00 | 35.17 | 55.81 | 42.81 | 62.67 |
| | IMP [75] | 24.21 | 39.79 | 38.68 | 59.16 | 47.08 | 66.11 |
| | LightGlue [32] | 21.64 | 35.98 | 38.27 | 58.91 | 47.75 | 66.75 |
| | DiffGlue (Ours) | **27.39** | **44.39** | **39.94** | **60.33** | **48.79** | **67.65** |
| ALIKED [79] | MNN | 0.08 | 0.37 | 32.34 | 52.32 | 32.72 | 51.72 |
| | LightGlue | 27.94 | 44.18 | 43.89 | 63.80 | 49.89 | 68.33 |
| | DiffGlue (Ours) | **35.37** | **53.18** | **44.55** | **64.39** | **51.86** | **69.78** |
| LoFTR [60] | | 5.24 | 13.24 | **39.80** | **60.03** | **42.64** | **62.08** |
| PDC-Net+ [63] | | **20.23** | **33.18** | 36.47 | 56.91 | 41.88 | 61.02 |

longest dimension is equal to 1600 pixels, then detect up to 2048 feature points with both SP [13] and ALIKED [79] per image as the input of image feature matching methods. We choose DLT, RANSAC [15] or MAGSAC++ [4] as a geometric model estimator. The **AUC** of the maxima error of rotation and translation at different thresholds (5° and 10°) is reported. We choose the same comparative methods as the homography estimation task. Additionally, following the settings in [64], we add the dense matching methods LoFTR [60] and PDC-Net+ [63] on YFCC100M dataset for further comparisons, where all images are resized so that their shortest dimension is equal to 480 pixels. All results are shown in Tables 2 and 3, DiffGlue consistently outperforms all other methods, revealing the fact that this diffusion-based pipeline allows the model to converge to better results, and that the prior learned by the Diffusion Model does guide the effective information propagation in the GNN. We also illustrate the qualitative results of feature matching and relative pose estimation in Figure 4. We use the eipolar error [19] to determine the correctness of a correspondence, considering the one with an error greater than 0.005 to be wrong, labeled in red, and with a smaller error labeled closer to green. In the top left corner, we indicate the algorithm (*i.e.*, SuperGlue, LightGlue, DiffGlue), the

rotation and translation errors with RANSAC (*i.e.*, $\Delta R$ and $\Delta t$), and the percentage of correct matches.

## 5.3 Visual Localization

Advances in feature matching can facilitate practical issues such as long-term visual localization [49, 51], which aims to recover the 6 degree-of-freedom (6-DOF) camera pose from a query image relative to a known 3D scene model. This task is greatly challenged in practice by a variety of complicated conditions such as viewpoint or illumination changes, thus an accurate matching method is required. Following [9, 32], we integrate different matching methods into the official Hloc [49] pipeline, evaluate them on both Aachen Day-Night v1.0 [51, 52] and InLoc [61]. Specifically, based on COLMAP toolbox [55], we first triangulate a 3D point cloud from all reference images with known poses and calibration, then retrieve 20 reference images for each query image with NetVLAD [1] on Aachen Day-Night v1.0 and 40 reference images on InLoc, matching the query image and the retrieved ones with image feature matching methods, where the feature points are detected up to 4096 by SP [13].

**Table 4: Visual localization on Aachen Day-Night v1.0 [51].**

| Feature+Matcher | | Day | Night |
|---|---|---|---|
| | | (0.25m, 2°) / (0.5m, 5°) / (5.0m, 10°) | |
| SP [13] | MNN | 86.9 / 92.0 / 95.5 | 73.5 / 79.6 / 88.8 |
| | MNN+ConvMatch [78] | 88.1 / 94.4 / 97.3 | 79.6 / 88.8 / 96.9 |
| | SuperGlue [50] | 87.9 / 95.0 / **97.9** | 84.7 / 92.9 / **99.0** |
| | SGMNet [9] | 86.5 / 93.7 / 97.2 | 82.7 / 91.8 / **99.0** |
| | ResMatch [12] | 86.8 / 93.7 / 97.2 | 81.6 / 91.8 / 98.0 |
| | LightGlue [32] | 88.0 / 93.8 / 97.5 | 84.7 / 91.8 / **99.0** |
| | DiffGlue (Ours) | **88.3** / **95.3** / 97.8 | **85.7** / **93.9** / 99.0 |
| COTR [24] | | 82.4 / 91.9 / 96.8 | 75.5 / 90.8 / 99.0 |
| LoFTR [60] | | **83.9** / **92.6** / **97.2** | 79.6 / **91.8** / **100.0** |

**Table 5: Visual localization on InLoc [61].**

| Feature+Matcher | | DUC1 | DUC2 |
|---|---|---|---|
| | | (0.25m, 10°) / (0.5m, 10°) / (1.0m, 10°) | |
| SP [13] | MNN | 30.3 / 48.5 / 57.1 | 23.7 / 38.2 / 45.0 |
| | SuperGlue [50] | 44.9 / 66.2 / **78.8** | 46.6 / **74.0** / **77.1** |
| | SGMNet [9] | 39.9 / 56.6 / 70.2 | 39.7 / 59.5 / 65.6 |
| | ResMatch [12] | 42.9 / 61.6 / 73.7 | 38.2 / 62.6 / 69.5 |
| | LightGlue [32] | 44.4 / 64.1 / 75.8 | 42.7 / 67.9 / 73.3 |
| | DiffGlue (Ours) | **46.5** / **67.2** / 78.3 | **49.6** / 71.8 / 76.3 |

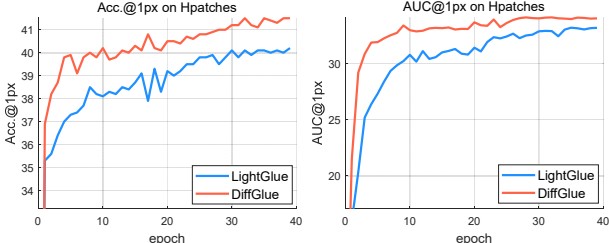

**Figure 5: Curves on Acc.@1px and AUC@1px.**

Finally, a camera pose can be estimated by RANSAC [15] and a Perspective-n-Point solver. We report the pose recall at multiple distance and orientation thresholds (*i.e.*, (0.25m, 2°), (0.5m, 5°), (5.0m, 10°)). We choose similar comparisons as relative pose estimation. For Aachen Day-Night v1.0, we further add dense matching methods LoFTR [60] and COTR [24], together with a state-of-the-art outlier rejection method ConvMatch [78]. Tables 4 and 5 show the promising performance of DiffGlue on visual localization.

## 5.4 Analysis

We further analyze DiffGlue in this section. First, we demonstrate the fast and accurate convergence property of DiffGlue. Then, we check the computational usage of DiffGlue to show its efficiency. We also conduct ablation studies to reveal the important roles of the Diffusion Model and Assignment-Guided Attention.

*5.4.1 Fast and Accurate Convergence.* We plot the curves of Light-Glue [32] and DiffGlue on **Acc.**@1px and **AUC**@1px estimated by DLT for HPatches [2] during the first training stage in Figure 5. DiffGlue converges faster and receives better performance than LightGlue, which benefits from the Diffusion Model and the injected

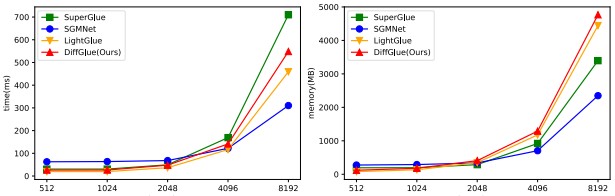

**Figure 6: Statistics of runtime and memory usage.**

**Table 6: Ablation studies.**

| Num. | Diff. | A.Att. | DLT | | RANSAC [15] | |
|---|---|---|---|---|---|---|
| | | | @5° | @10° | @5° | @10° |
| (a) | ✓ | ✓ | **43.21** | **59.13** | **50.21** | **67.30** |
| (b) | ✓ | | 41.06 | 56.96 | 49.11 | 66.29 |
| (c) | | ✓ | 40.44 | 56.16 | 49.65 | 66.51 |
| (d) | | | 38.78 | 54.39 | 47.34 | 65.46 |

correspondence prior, allowing the network to better converge and avoid falling into local optimum during training.

*5.4.2 Computational Usage.* We test the runtime and memory requirements of SuperGlue [50], SGMNet [9], LightGlue and DiffGlue *w.r.t.* the increasing numbers of input, where the input feature points are randomly generated. Statistic results are illustrated in Figure 6. It can be seen that DiffGlue is able to maintain real-time nature and competitive computing resource consumption while achieving optimal performance on multiple tasks.

*5.4.3 Ablation Studies.* We conduct ablation studies by repeating the relative pose estimation on MegaDepth-1500 [30], and report **AUC** with DLT and RANSAC [15] in Table 6. (**a**) is the full DiffGlue with both the Diffusion Model as a framework (note as Diff.) and Assignment-Guided Attention (note as A.Att.) as the prior injection module. (**b**) eliminates the Assignment-Guided Attention only. (**c**) removes the diffusion-based framework so that the prior assignment matrix used in the Assignment-Guided Attention of the $\ell$-th layer comes from $\tilde{P}^{\ell-1}$ outputted by the former layer as seen in Eq. (29), and we define $\tilde{P}^0 = \mathcal{S}(F_A^0 (F_B^0)^T)$ refer to Eq. (7). (**d**) is a baseline that eliminates both components. Table 6 reveals that DiffGlue benefits from all the ingredients mentioned above.

## 6 Conclusion

We design a novel framework called DiffGlue that introduces the Diffusion Model into the sparse image feature matching problem. With the inherent prior in the Diffusion Model and the wider exploration of the solution space by stochastic Gaussian noise, DiffGlue will search for the direction of gradient descent in the complex assignment matrix so that learning a suitable optimization path, leads the network training step by step, and finally avoids falling into a local optimum. Additionally, Assignment-Guided Attention is proposed to inject the correspondence prior into the denoising network, enhancing the information propagation in the GNN and making merging the Diffusion Model into the sparse matching pipeline possible, resulting in faster and more accurate convergence during training. Extensive experiments demonstrate the superiority of our method and the promising properties mentioned above.

# Acknowledgments

This work was supported by National Natural Science Foundation of China (No. 62276192).

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
