# OpenReview forum: "DiffGlue: Diffusion-Aided Image Feature Matching"
_acmmm.org/ACMMM/2024/Conference — MM2024 Poster_

### Official Review · Reviewer_FCdJ · 2024-05-22

**Rating:** 3
**Confidence:** 3

**Summary:**

The paper introduces a novel method for image feature matching, a fundamental problem in computer vision. The authors propose incorporating the Diffusion Model into the feature matching framework, which allows for an incremental approach to finding the global optimal solution in the assignment matrix space. The method, DiffGlue, utilizes a special Assignment-Guided Attention mechanism to integrate the Diffusion Model with sparse image feature matching, enhancing the information propagation within the Graph Neural Network (GNN). The paper demonstrates DiffGlue's effectiveness through extensive experiments on 3 tasks ( homography estimation, relative pose estimation, and visual localization ), where it outperforms other methods. The ablation studies show the effectiveness of the diffusion mechanism and assignment-guided attention mechanism.

However, I think one important literature, DiffMatch (as shown in Limitations) is missing, which also uses Diffusion Model for image feature matching. Without comparing with it, I can only give my rating borderline reject. I hope the authors can address this issue in the rebuttal. If the authors can show that they do outperform DiffMatch and address my concerns in limitations, I will improve my rating.

**Strengths:**

1. It is one of the first work to apply diffusion model for this task. The integration of the Diffusion Model into image feature matching is a novel contribution that offers a new perspective on solving the correspondence problem; The proposed Assignment-Guided Attention mechanism serves as an effective bridge between the Diffusion Model and GNN, potentially improving information flow and matching quality.
2. The paper provides extensive experimental results, demonstrating the method's superiority over existing state-of-the-art techniques. The experiments setting is the same as previous work to ensure a fair comparison.
3. The paper is well-structured, with a clear presentation of the problem, the proposed solution, and the experimental results. The figures and tables are informative and support the text effectively.

**Limitations:**

1. The paper could benefit from a more detailed comparison with other contemporary models like DiffMatch [1]. It will be better for the proposed method to also test on optical flow estimation task and make a comparison with [2]
2. The authors claim that their method "for the first time introduces the Diffusion Model into the image feature matching framework". However, considering the existence of DiffMatch[1], this might be overclaimed.
3. The authors frequently mention that the Diffusion Model divides the optimization path into several parts during training instead of the common path with a direct gradient descent, achieving a better convergence and the global optimum. Although a clear figure is provided to demonstrate this, the authors have not explained the underlying reasons nor have they cited appropriate references to support it. Providing clarification on this point would facilitate a better understanding for readers who are not familiar with the Diffusion theory.

[1] DiffMatch: Diffusion Model for Dense Matching (ICLR'24 Oral)

[2] The surprising effectiveness of diffusion models for optical flow and monocular depth estimation. Advances in Neural Information Processing Systems 36 (2024).

**Suitability:**

3

---

### Official Review · Reviewer_9DNr · 2024-05-25

**Rating:** 4
**Confidence:** 1

**Summary:**

The paper proposes an innovative approach to address the challenge of predicting assignment matrices using direct mapping. By drawing inspiration from the gradual diffusion process in image denoising, the authors suggest optimizing the assignment step by step, with each prediction serving as a prior for the next step.

**Strengths:**

This concept is effectively translated into the training of a feature matching network within the diffusion framework, which is clearly elucidated in the paper. Each component is clearly explained. The paper is well-organized and easy to follow. The experiments on homography estimation and relative pose estimation provide compelling evidence of the effectiveness of this diffusion model compared to other feature matching methods. Despite not being well-experienced in feature matching techniques, I find the achieved performance metrics such as accuracy and AUC to be solid.

**Limitations:**

while the paper illustrates the step-by-step training strategy and the utilization of prior information from previous estimations, I believe there is a lack of detailed analysis or empirical demonstration regarding how these aspects help avoid local minima, beyond the provided illustration. It would be beneficial to have concrete evidence showcasing the importance of these claims.

Additionally, the paper mentions the challenge posed by sparse feature points in feature matching, but there is a lack of analysis on this aspect and how it is addressed within the diffusion framework. It would be insightful to delve into the specific difficulties encountered and the strategies employed to mitigate them.

Furthermore, Table 4 and 5, while presented in the paper, lack clear explanations. Providing detailed descriptions or annotations for these tables would enhance the reader's understanding of the results presented.

**Suitability:**

2

---

### Official Review · Reviewer_ce3B · 2024-05-27

**Rating:** 4
**Confidence:** 3

**Summary:**

The article proposes a novel image feature matching method called DiffGlue, which can achieve fast and accurate matching results in multiple vision tasks. DiffGlue is based on iterative diffusion and denoising processes, and can be guided step by step by the prior of the diffusion model during training to progressively approach the optimal solution. In addition, DiffGlue includes a special "Assignment-Guided Attention" that serves as a bridge connecting the diffusion model and sparse image feature matching, injecting inherent priors into the GNN to improve information transfer. Extensive experiments have shown that DiffGlue converges faster and better, outperforming state-of-the-art technologies in applications such as homography estimation, relative pose estimation, and visual localization.

**Strengths:**

1.	For the first time, the diffusion model is applied to the problem of image feature matching, utilizing its ability to establish mappings in complex spaces, assisting the model in progressively approaching the global optimal solution during the training process.
2.	By introducing Gaussian noise in stages during training and gradually removing it ( the diffusion and denoising process), the model can more effectively search for the direction of gradient descent in the complex assignment matrix space, accelerating convergence and avoiding getting stuck in local optima.
3.	A novel attention mechanism is proposed, which uses the correspondence predicted by the diffusion model as a prior, injecting it into the graph neural network to improve information propagation and enhance the accuracy of matching.

**Limitations:**

1.Diffusion models typically involve multiple steps of iterative processes. It is necessary to evaluate whether DiffGlue has a higher computational complexity compared to other methods, and whether this would affect its runtime.
2.DiffGlue shows excellent performance on specific datasets (such as MegaDepth, HPatches, etc.), but how is its ability to generalize?

**Suitability:**

3

---

### Meta-Review · Area_Chair_kQ5F · 2024-07-05

**Recommendation:** Accept (Poster)
**Confidence:** 5

**Metareview:**

The manuscript presents a novel application of the Diffusion Model for image feature matching, integrating it with Graph Neural Networks through an innovative Assignment-Guided Attention mechanism. The extensive experimental results demonstrate its superiority over existing techniques. While the manuscript is well-structured and provides clear illustrations, it would benefit from a more detailed comparison with contemporary models like DiffMatch, a thorough analysis of its training strategy's effectiveness in avoiding local minima, and clear explanations for some tables. Additionally, evaluating the computational complexity and generalization ability would enhance its robustness. The authors provide a decent rebuttal to each of these concerns. All three reviewers recommended to accept this paper for publication. The authors are encouraged to fully address the reviewers' comments in their final version.